# Canopy Size and Light Use Efficiency Explain Growth Differences between Lettuce and Mizuna in Vertical Farms

**DOI:** 10.3390/plants10040704

**Published:** 2021-04-06

**Authors:** Theekshana C. Jayalath, Marc W. van Iersel

**Affiliations:** Department of Horticulture, University of Georgia, Athens, GA 30602, USA; tcjay@uga.edu

**Keywords:** canopy size, incident light, light interception, light use efficiency, mizuna, projected canopy size, quantum yield of photosystem II

## Abstract

Vertical farming is increasingly popular due to high yields obtained from a small land area. However, the energy cost associated with lighting of vertical farms is high. To reduce this cost, more energy efficient (biomass/energy use) crops are required. To understand how efficiently crops use light energy to produce biomass, we determined the morphological and physiological differences between mizuna (*Brassica rapa* var. *japonica*) and lettuce (*Lactuca sativa* ‘Green Salad Bowl’). To do so, we measured the projected canopy size (PCS, a morphological measure) of the plants throughout the growing cycle to determine the total amount of incident light the plants received. Total incident light was used together with the final dry weight to calculate the light use efficiency (LUE, g of dry weight/mol of incident light), a physiological measure. Plants were grown under six photosynthetic photon flux densities (PPFD), from 50 to 425 µmol m^−2^ s^−1^, for 16 h d^−1^. Mizuna and lettuce were harvested 27 and 28 days after seeding, respectively. Mizuna had greater dry weight than lettuce (*p* < 0.0001), especially at higher PPFDs (PPFD ≥ 125 µmol m^−2^ s^−1^), partly because of differences in the projected canopy size (PCS). Mizuna had greater PCS than lettuce at PPFDs ≥ 125 µmol m^−2^ s^−1^ and therefore, the total incident light over the growing period was also greater. Mizuna also had a higher LUE than lettuce at all six PPFDs. This difference in LUE was associated with higher chlorophyll content index and higher quantum yield of photosystem II in mizuna. The combined effects of these two factors resulted in higher photosynthetic rates in mizuna than in lettuce (*p* = 0.01). In conclusion, the faster growth of mizuna is the result of both a larger PCS and higher LUE compared to lettuce. Understanding the basic determinants of crop growth is important when screening for rapidly growing crops and increasing the efficiency of vertical farms.

## 1. Introduction

Vertical farming refers to hydroponic crop production in buildings with precise environmental control. Vertical farms do not require arable land and can obtain high crop yields from a small land area. Therefore, vertical farms are becoming popular in urban areas. However, the energy costs associated with the required electric lighting and environmental control, especially cooling and dehumidification, in vertical farming are high [1]. The cost of electric lighting in controlled environment agriculture in just the United States has been estimated at ~$600 million annually [2]. To bring this cost down, research into more efficient production techniques [3] and more energy-efficient crops (more biomass gain per unit of energy use) are required.

Overall, crop growth is a function of the amount of incident light reaching the canopy, which depends on projected canopy size (PCS) [4,5], and light use efficiency (LUE, grams of biomass produced per mol of incident light) [6]. To screen for crops with rapid growth, quantifying PCS development and LUE is essential.

Plants that produce a larger canopy can achieve faster growth by increasing the amount of incident light reaching the canopy compared to plants with a smaller canopy [4,5]. As incident light increases, canopy photosynthesis and biomass accumulation of plants also increase [5], as long as canopy photosynthesis is not light-saturated. This in turn helps plants to produce additional canopy faster than plants with a smaller canopy [4]. Therefore, quantifying the PCS and determining how this affects the amount of incident light, is important. Non-destructive digital imaging has been used in many crops, including tomatoes [7], soybean [8], and lettuce [5,9]. Periodic PCS measurements can be used to estimate the daily PCS [6]. Combining those data with PPFD data allows for estimation of the total incident light over the course of the growing cycle. Projected canopy size is also valuable to make crop growth predictions, since PCS early in the growing cycle may be correlated with the final dry weight of the crop, as we have previously shown in lettuce (*Lactuca sativa*) [6,10] and black-eyed susan (*Rudbeckia fulgida*) [11].

The LUE of a crop describes how efficiently plants use the incident light for growth. The LUE can be calculated by dividing the dry weight of a plant by the total incident light that the plant received throughout the growing period [6]. This provides a physiological measure of how efficiently crops use light, in contrast to calculating LUE based on the amount of light provided to the growing space, which provides insight into production efficiency, but not underlying physiological mechanisms [6]. Factors such as chlorophyll content, the quantum yield of photosystem II (Φ_PSII_, the fraction of absorbed photons used to drive photochemistry), and CO_2_ assimilation are all important in determining the LUE of a crop.

Higher chlorophyll content increases light absorption [12]. The energy of the absorbed photons can then be used in the light reactions of photosynthesis to drive photochemistry (electron transport), while some light energy is dissipated as heat (NPQ, non-photochemical quenching), or re-emitted as fluorescence. Photochemistry competes with NPQ and chlorophyll fluorescence for excitation energy from photons [13,14]. To understand how different lighting strategies affect plant growth, it is important to determine Φ_PSII_, since this partly determines how efficiently plants use the absorbed light to drive photosynthesis and produce biomass. Previous studies have shown that increasing PPFD decreases Φ_PSII_, due to the partial closing of photosystem II reaction centers and upregulation of NPQ [15,16]. Although Φ_PSII_ decreases, the electron transport rate (ETR) increases with increasing PPFD [15,16,17]. The Φ_PSII_ of different species responds differently to increasing PPFDs [16]. In addition, plants acclimated to high light have higher Φ_PSII_ and ETR than those acclimated to low light. Therefore, to increase the production efficiency in plant factories, identifying crops with both high Φ_PSII_ and high ETR is important, because a higher ETR will result in the production of more ATP and NADPH for use in CO_2_ assimilation. Both high Φ_PSII_ and high chlorophyll content index (CCI) may increase the ETR, CO_2_ assimilation rate, and potentially the growth rate of plants.

A previous indoor study with mizuna (*Brassica rapa* var. *japonica*) and oakleaf lettuce (*Lactuca sativa* ‘Green Salad Bowl’) found much faster growth of mizuna compared to lettuce [18]. This growth difference must be the result of morphological and/or physiological differences between the two crops, but those underlying reasons were not explored in that study. A better understanding of the underlying reasons for growth differences among crops will facilitate screening for rapid-growing crops and cultivars that are well-suited for vertical farming production systems. It can also enable breeding efforts by providing selection criteria for crops that are well-suited for production in vertical farms. Therefore, our objective was to determine the underlying mechanisms for the growth differences between mizuna and lettuce. We hypothesize that the faster growth of mizuna is the result of a greater PCS, increased canopy incident light, and higher LUE. Since crop growth is affected by PPFD, we also determined how different PPFDs affect the morphological and physiological factors underlying crop growth. Comparing two species, grown at different PPFDs, allowed us to determine how useful PCS and LUE are in explaining crop growth.

## 2. Results

### 2.1. Experiment

To determine the underlying morphological and/or physiological reasons for growth differences between mizuna and lettuce at different PPFDs, plants were grown at six PPFD levels (~50, 125, 200, 275, 350, and 425 µmol m^−2^ s^−1^ at the center of each section) for a 16-hr photoperiod. Projected canopy size was measured twice a week throughout the growing period, and those images were used to estimate the daily PCS and to calculate the total incident light per plant over the growing period. In addition, leaf chlorophyll content index (CCI), anthocyanin content index (ACI), Φ_PSII_, and net CO_2_ assimilation of both crops were measured during the study. Mizuna and lettuce were harvested 27 and 28 days after seeding, respectively. The total leaf area and shoot dry weight were measured. Finally, the LUE was calculated by dividing shoot dry weight by the total incident light over the growing period.

### 2.2. Projected Canopy Size

Projected canopy size of both crops increased sigmoidally over time. A PPFD of 50 µmol m^−2^ s^−1^ resulted in a much lower PCS of both crops compared to PPFD levels ≥ 125 µmol m^−2^ s^−1^. The PCS of mizuna at the end of the growing cycle was ~340 cm^2^/plant at PPFDs ≥ 200 µmol m^−2^ s^−1^ (Figure 1 and Figure 2). For lettuce, PCS was similar (~240 cm^2^/plant) at all PPFDs ≥ 125 µmol m^−2^ s^−1^ (Figure 1 and Figure 2). Mizuna had a larger PCS than lettuce starting from the early growth stages at PPFDs > 50 µmol m^−2^ s^−1^ (Figure 1). The difference in PCS between the two species increased over time, especially at higher PPFDs (≥ 200 µmol m^−2^ s^−1^) (Figure 1).

The PCS of both crops at harvest was low at a PPFD of 50 µmol m^−2^ s^−1^ and increased asymptotically with increasing PPFD (*p* < 0.0001, Figure 2). The PCS of both crops was similar at PPFDs of 50 and 125 µmol m^−2^ s^−1^. However, the PCS of lettuce did not increase further at PPFDs ≥ 125 µmol m^−2^ s^−1,^ while mizuna PCS was similar at PPFDs ≥ 200 µmol m^−2^ s^−1^. At PPFDs ≥ 200 µmol m^−2^ s^−1^, mizuna had a greater PCS than lettuce (Figure 2).

### 2.3. Incident Light

The incident light integrated over the entire crop cycle increased with higher PPFD levels for both crops, but this increase was more pronounced for mizuna than for lettuce (Figure 3, *p* < 0.0001).

### 2.4. Chlorophyll Content Index and Anthocyanin Content Index

Increasing the PPFD increased both CCI and anthocyanin content index (ACI) of both crops (*p* ≤ 0.003) (Figure 4). This was more pronounced in mizuna than in lettuce; CCI of mizuna increased by 0.08 for each µmol m^−2^ s^−1^ increase in PPFD, compared to an increase of 0.01 per µmol m^−2^ s^−1^ in lettuce. As the PPFD increased from 50 to 400 µmol m^−2^ s^−1^, the CCI of mizuna increased from ~7 to ~40, while that of lettuce increased from ~2 to about ~10 (Figure 4A).

The ACI showed a similar pattern as CCI; increasing PPFD increased ACI in both crops (*p* ≤ 0.003) (Figure 4B). Mizuna had a higher ACI than lettuce at all PPFD levels (*p* < 0.0001) and mizuna ACI increased more rapidly with increasing PPFD than that of lettuce. For each 1 µmol m^−2^ s^−1^ increase in PPFD, the ACI of mizuna increased by 0.015 and that of lettuce by 0.004. As the PPFD increased from 50 to 400 µmol m^−2^ s^−1^, the ACI of mizuna increased from ~3 to ~8, while that of lettuce only increased from ~2 to about ~3.

### 2.5. Quantum Yield of Photosystem II and CO_2_ Assimilation

The quantum yield of photosystem II (Φ_PSII_) of both crops decreased linearly with increasing PPFD (*p* = 0.0008) (Figure 5A). Increasing PPFD by 1 µmol m^−2^ s^−1^ reduced Φ_PSII_ of lettuce and mizuna by 0.0003 mol mol^−1^. Mizuna always had a higher Φ_PSII_ (~0.05 mol mol^−1^) than lettuce regardless of PPFD (*p* < 0.0001). The net CO_2_ assimilation rate of both crops increased with increasing PPFD, but this tended to be more pronounced in mizuna than in lettuce (*p* = 0.08). Both crops had a CO_2_ assimilation rate of ~1 µmol m^−2^ s^−1^ at a PPFD of 50 µmol m^−2^ s^−1^, but at PPFD of 425 µmol m^−2^ s^−1^ mizuna had a CO_2_ assimilation rate of ~18 µmol m^−2^ s^−1^ while that of lettuce was only ~13 µmol m^−2^ s^−1^. The assimilation rate of mizuna and lettuce increased by 0.044 and 0.035 µmol m^−2^ s^−1^ per 1 µmol m^−2^ s^−1^ increase in PPFD, respectively.

### 2.6. Final Leaf Area and Canopy Overlap Ratio

The final leaf area of both mizuna and lettuce increased asymptotically with increasing PPFD (*p* < 0.0001) (Figure 6A). Both crops had the highest leaf area at PPFDs ≥ 275 µmol m^−2^ s^−1^, but lettuce leaf area increased faster with increasing PPFD than mizuna leaf area (*p* < 0.0001). At a PPFD of ≥ 275 µmol m^−2^ s^−1^ lettuce had a leaf area of ~1200 cm^2^ per plant, while that of mizuna was only ~800 cm^2^. However, lettuce had a lower PCS at harvest compared to mizuna at PPFDs ≥ 200 µmol m^−2^ s^−1^ (Figure 2). This apparent contradiction between a greater leaf area and a lower PCS of lettuce can be explained by the canopy overlap ratio (leaf area/PCS); lettuce had a much higher canopy overlap ratio than mizuna at PPFDs ≥ 125 µmol m^−2^ s^−1^ (Figure 6B). The canopy overlap ratio of lettuce increased more rapidly, from 1.2 to 5.2 with increasing PPFD compared to that of mizuna (increasing from 1.1 to 2.3) (*p* < 0.0001).

### 2.7. Shoot Dry Weight and Specific Leaf Area

The shoot dry weight at 425 µmol m^−2^ s^−1^ was ~50 times higher than at 50 µmol m^−2^ s^−1^ for both lettuce and mizuna, although PPFD increased only ~9 × (Figure 7A, Appendix A). Lettuce and mizuna had a dry weight of 0.08 and 0.11 g/plant at a PPFD of 50 µmol m^−2^ s^−1^ and 3.85 and 6.02 g/plant at a PPFD of 425 µmol m^−2^ s^−1^, respectively. A 1 µmol m^−2^ s^−1^ increase in PPFD increased the dry weight of lettuce by 10.0 mg/plant and that of mizuna by 15.8 mg/plant. With increasing PPFD, the specific leaf area (SLA, leaf area per gram of dry weight) of both crops decreased (*p* < 0.0001) (Figure 7B). However, due to the higher leaf area and lower dry weight of lettuce compared to mizuna, the SLA of lettuce was greater than that of mizuna at all PPFDs. The difference in SLA between the two species decreased with increasing PPFD (Figure 7B).

### 2.8. Light Use Efficiency

Mizuna had a higher LUE than lettuce (*p* < 0.0001) (Figure 8). The LUE of mizuna was ~1.1 g mol^−1^ at PPFDs up to 200 µmol m^−2^ s^−1^ and decreased to ~0.75 g mol^−1^ at a PPFD of 425 µmol m^−2^ s^−1^. In contrast, lettuce LUE was greatest at PPFDs of 125 to 350 µmol m^−2^ s^−1^ (~0.8 g mol^−1^) and ~0.6 g mol^−1^ at PPFDs of 50 and 425 µmol m^−2^ s^−1^.

## 3. Discussion

Mizuna grew faster than lettuce at PPFDs ≥ 125 µmol m^−2^ s^−1^ (Figure 7A). A previous study conducted to identify the effects of photoperiod on leafy greens found much faster growth of mizuna compared to lettuce [18]. The difference in biomass between the two crops increased with increasing PPFD (and shorter photoperiods). The goal of the current study was to determine the underlying reasons for the growth differences between these two crops.

### 3.1. Projected Canopy Size and Incident Light

Mizuna had a larger PCS than lettuce, starting from the early growth stages (Figure 1; Appendix A). One reason for this early difference in PCS is the faster germination and larger cotyledons of mizuna compared to lettuce. Due to their small canopy size, seedlings capture only a small fraction of the provided light. Therefore, increased PCS at early stages can increase light capture and growth of seedlings [11]. A previous study compared different lettuce cultivars and found that early PCS was a good predictor of final shoot biomass [10]. We observed the same pattern in our study, with a strong positive correlation (*R* = 0.91 for lettuce and *R* = 0.89 for mizuna, *p* < 0.0001) between early PCS (lettuce 10 d and mizuna 8 d after seeding) and the dry weight of both lettuce and mizuna (Appendix A).

This higher PCS of mizuna during the early part of the growing cycle (at PPFDs ≥ 125 µmol m^−2^ s^−1^) may have helped it to capture more light and grow faster than lettuce. A previous greenhouse study conducted with ‘Little Gem’ lettuce observed the same trend; plants with larger PCS in early growth stages absorbed more light, grew faster, and produced additional canopy faster than plants with a smaller PCS [4]. In that prior study, plants were grown with different photoperiods, but the same DLI. In contrast, plants in the current study were grown with the same photoperiod, but different PPFDs, thus resulting in different DLIs. 

At a PPFD of 50 µmol m^−2^ s^−1^, both mizuna and lettuce grew slowly and had a low PCS at harvest compared to plants grown at higher PPFDs (Figure 2). At PPFDs ≥ 125 µmol m^−2^ s^−1^, the PCS of both crops was much greater than at a PPFD of 50 µmol m^−2^ s^−1^. However, at PPFD ≥ 200 µmol m^−2^ s^−1^, the PCS of mizuna was greater than that of lettuce (Figure 2) (*p* < 0.0001). The growth difference between the two crops is at least partly because of the observed differences in PCS and its impact on incident light. The incident light of mizuna increased more rapidly with increasing PPFD than that of lettuce, consistent with the increasing difference in dry weight between the two crops as PPFD increased (Figure 3 and Figure 7). Other studies also mentioned a positive correlation between PCS and incident light [5,11,19]. Additionally, several other studies on leafy greens show a positive correlation between biomass gain and incident light integrated over the entire crop cycle [5,20,21]. 

### 3.2. Leaf Area, Specific Leaf Area, and Canopy Overlap Ratio

Even though lettuce had a lower PCS than mizuna, its total leaf area was larger than that of mizuna at PPFDs ≥ 125 µmol m^−2^ s^−1^ (Figure 6A). This was due to the differences in leaf arrangement and morphology between the two species. The canopy overlap ratio of lettuce was higher than that of mizuna at PPFDs ≥ 125 µmol m^−2^ s^−1^ (Figure 6B), resulting in more intra-canopy shading in lettuce. With increasing PPFD, the differences in leaf area and canopy overlap ratio between the two species increased.

The SLA of lettuce was greater compared to mizuna, due to a larger total leaf area and lower shoot dry weight than mizuna at PPFDs ≥ 125 µmol m^−2^ s^−1^ (Figure 7B).

### 3.3. Chlorophyll Content Index and Anthocyanin Content Index

We observed a higher CCI in mizuna leaves compared to lettuce at all PPFDs (Figure 4A). Leaf light absorptance is positively associated with the CCI [12]. Therefore, mizuna likely had a higher leaf light absorptance in lettuce. However, we do not have the actual light absorptance data for this study.

With increasing PPFD, CCI increased in both species (Figure 4A). In previous studies, CCI increased with lower PPFD [4,18]. In those studies, lower PPFDs were combined with longer photoperiods to maintain the same DLI. Chlorophyll production is a light-regulated process [22]. Therefore, the CCI increase in low PPFD treatments in prior studies was associated with longer photoperiods, which increases the amount of time available for plants to produce chlorophyll [18]. In our study, photoperiod was the same in all treatments and therefore the daily light integral was higher at higher PPFDs.

In response to increasing PPFD, the CCI increase in mizuna was about four times greater than in lettuce, indicating that mizuna acclimates more strongly to different PPFDs than lettuce (Figure 4A). The greater CCI increase in mizuna may be partly the result of a lower SLA at PPFDs ≥ 125 µmol m^−2^ s^−1^, compared to lettuce (Figure 7B). A higher SLA is associated with thinner leaves, which typically have low chlorophyll content per unit leaf area [23]. In our study, increasing PPFD decreased the SLA of both crops (Figure 7B). Such a decrease in SLA can increase the CCI, due to increased leaf thickness. We indeed observed strong negative correlations between the SLA and CCI of mizuna (*R* = −0.76, *p* = 0.0003) and lettuce (*R* = −0.79, *p* = 0.0001; Appendix A), but this relationship differed greatly between the two species. Mizuna’s CCI decreased much more quickly with increasing SLA than that of lettuce. At mizuna’s highest SLA (~375 cm^2^ g^−1^), its CCI was similar (~5) to the lettuce CCI at its lowest SLA (~325 cm^2^ g^−1^)

The anthocyanin content index (ACI) was measured to identify whether the two crops differ in anthocyanin accumulation in response to increasing PPFD. Anthocyanins in leaves have a protective role against intense light and help dissipate excess excitation energy [24,25]. The ACI of both species did indeed increase with increasing PPFD (Figure 4B). However, the ACI of both crops was much lower than those previously reported for red leaf basil (ACI of 28–81) [26] and pak choi (ACI of 35–65) [27]. Consistent with the low ACI, we did not observe any red coloration on leaves of mizuna or lettuce, with increasing PPFD.

### 3.4. Quantum Yield of Photosystem II and CO_2_ Assimilation

With increasing PPFD, Φ_PSII_ of both crops decreased at a similar rate (Figure 5A). At higher PPFD, a larger proportion of the PSII reaction centers are closed and more of the absorbed light energy is dissipated as heat to minimize photoinhibition [15,28]. This rise in heat dissipation results in a smaller fraction of the excitation energy being directed towards the PSII reaction centers, reducing Φ_PSII_ [17,29]. Many previous studies have observed a decrease in Φ_PSII_, but increasing ETR, in response to increasing PPFD [15,16,17]. We observed the same pattern in Φ_PSII_ with increasing PPFD. We did not calculate the ETR because leaf absorptance was not measured. However, the higher CCI of mizuna suggests higher leaf absorptance compared to lettuce [12], in which case differences in electron transport rate between the two species would have been larger than the differences in Φ_PSII_. Even though we observed a reduction in Φ_PSII_ of both crops with increasing PPFD, the Φ_PSII_ of mizuna was always higher (~0.05 mol mol^−1^) than that of lettuce (Figure 5A) (*p* < 0.0001). A study conducted to understand the effects of different PPFDs on the photochemistry of three species adapted to different light levels, found a higher Φ_PSII_ and higher ETR in high-light adapted species compared to the species adapted to moderate or low-light [16]. This suggests that mizuna is better adapted to high light levels than lettuce, and therefore has a higher Φ_PSII_.

With increasing PPFD, the net CO_2_ assimilation rate of mizuna tended to increase more rapidly than that of lettuce (Figure 5B, *p* = 0.08). This is consistent with the higher CCI and Φ_PSII_ of mizuna. The lower SLA of mizuna suggests thicker leaves compared to lettuce. The higher CCI associated with thicker leaves can increase the light absorptance and CO_2_ assimilation rate per unit leaf area [23].

### 3.5. Light Use Efficiency

Mizuna had a higher maximum (1.26 g mol^−1^) LUE than lettuce (0.74 g mol^−1^) (*p* < 0.0001) (Figure 8). Mizuna had its maximum LUE at PPFDs from 50 to 200 µmol m^−2^ s^−1^_,_ while lettuce LUE was maximal at PPFDs of 200 to 275 µmol m^−2^ s^−1^. This indicates that mizuna can convert incident light into biomass more efficiently than lettuce, especially at lower PPFDs. The maximum lettuce LUE we observed (0.74 g mol^−1^) is slightly higher than the LUE previously reported (0.61–0.65 g mol^−1^) for ‘Green Salad Bowl’ lettuce [6]. Legendre and van Iersel [6] used the same method to calculate LUE, but, did not use CO_2_ enrichment which may have reduced leaf photosynthesis and thus LUE.

Other studies reported substantially lower LUE values (<0.6 g mol^−1^) for lettuce [23,30], but in those studies, LUE calculations were based on the amount of light provided to the growing space, rather than light reaching the canopy of the crop. Their LUE values are thus heavily dependent on the plant density of the growing spaces. Light use efficiency can also vary among cultivars [31,32]. Therefore, it is hard to compare our LUE values of lettuce with those studies. There are no prior reports for mizuna LUE.

Multiple factors contributed to the higher LUE of mizuna. The combined effects of more chlorophyll and higher Φ_PSII_ of mizuna likely resulted in higher ETR and thus more photosynthesis than in lettuce. High photosynthetic rates can increase the relative growth rate, which in turn reduces the fraction of carbohydrates allocated to maintenance respiration and increases carbon use efficiency [33]. That in turn can increase LUE.

The LUE decreased at high PPFDs for both mizuna (PPFD ≥ 200 µmol m^−1^ s^−1^) and lettuce (PPFD ≥ 350 µmol m^−1^ s^−1^) (Figure 8). This reduction of LUE at high PPFD may be due to the decrease of Φ_PSII_ with increasing PPFD (Figure 5).

### 3.6. Conclusions

The Asian leafy green mizuna (*Brassica rapa* var. *japonica*) grows faster than oakleaf lettuce (*Lactuca sativa* ‘Green Salad Bowl’) when PPFD ≥ 125 µmol m^−2^ s^−1^. This faster growth of mizuna is the result of a higher CCI and larger PCS (at PPFDs ≥ 125 µmol m^−2^ s^−1^), allowing mizuna to capture more light, a higher Φ_PSII_ and net CO_2_ assimilation, and a higher LUE than lettuce. This study provides a framework for determining underlying morphological and physiological reasons for growth differences among crops. Understanding the basic determinants of crop growth is important to increase crop productivity and energy efficiency in vertical farms. Canopy imaging can be used to select crops that will grow well in vertical farms. Although we looked at a multitude of morphological and physiological factors, quantifying PCS and LUE would be adequate for the selection of crops with fast growth. Perhaps most intriguingly, our results confirm that early differences in PCS (8–10 days after seeding) are a good predictor of final biomass. Therefore, it may be possible to simply use early PCS to screen crops for rapid growth in controlled environment agriculture. This would allow for rapid throughput phenotyping and greatly accelerate the selection of promising genotypes.

## 4. Materials and Methods

### 4.1. Growth Chamber Setup

The study was conducted in a 4.4 m wide and 4.1 m long walk-in growth chamber. Cooling was provided using a top-mount refrigeration system and a dehumidifier maintained the relative humidity inside the growth chamber. The CO_2_ level inside the growth chamber was measured and maintained by triggering a solenoid valve to open and release CO_2_ from a compressed gas cylinder for 1-second intervals, whenever the CO_2_ concentration dropped below 800 µmol mol^−1^, using a CO_2_ transmitter (GMC20; Vaisala, Helsinki, Finland) and a datalogger (CR6, Campbell Scientific, Logan, UT, USA). Temperature and relative humidity measurements were collected every ten seconds with a probe (HMP50; Vaisala) connected to the datalogger. Using those temperature and relative humidity values, vapor pressure deficit (VPD) was calculated. Average temperature, CO_2_ level, and VPD inside the growth chamber were 24.2 ± 0.2 °C, 825 ± 38 µmol mol^−1^, and 1.4 ± 0.12 kPa (mean ± SD), respectively.

The growth chamber contained three 2.4 m long × 0.6 m wide × 2.2 m high metal shelving racks with 0.9 m distance between the racks. Each rack had three shelves with a 0.6 m × 2.4 m ebb-and-flow tray on each shelf. Each tray had an individual irrigation tube connected to a submersible pump. Those pumps were submerged in a fertigation tank located under the bottom shelf of each of the three metal racks. Three pumps were submerged in each fertigation tank. Each ebb-and-flow tray was divided into two 1.2 m long sections. Therefore, each rack had six 1.2 m long × 0.6 m wide × 0.6 m high sections, for a total of 18 growing sections. Each growing section had two 1.1 m-long white LED lights (RAY series with Physiospec indoor spectrum; Fluence Bioengineering, Austin, TX, USA) hanging 0.4 m above the bottom of the ebb & flow tray. Air circulation was provided by four 4 × 4 cm^2^ fans in each growing section.

### 4.2. Seeding and Plant Management

We placed two groups of nine 10-cm square pots in each growing section. Those pots were filled with a soilless substrate [80% peat: 20% perlite (v/v) (Fafard 1P; SunGro Horticulture, Agawam, MA, USA)]. Nine pots were seeded with mizuna (*Brassica rapa* var. *japonica*) and nine pots with lettuce (*Lactuca sativa* ‘Green Salad Bowl’). To prevent algae growth on the surface of the substrate, the top 1 cm of each pot was filled with calcined clay (Turface^®^ Pro League Elite, Profile Products LLC, Buffalo Grove, IL). Plants were sub-irrigated daily for 5 minutes with a nutrient solution containing 100 mg L^–1^ N made with a water-soluble fertilizer (15N–2.2P–12.45K, Peters Excel 15–5–15 Cal-Mag Special; Everris NA Inc, Dublin, OH, USA). Algaecide (ZeroTol 2.0, BioSafe Systems LLC, East Hartford, CT, USA) was applied to the surface of the substrate twice during the study, at a ratio of 1:400 (ZeroTol: water) as an algae preventative. Plants were grown under six treatments with different photosynthetic photon flux densities (PPFD) (~50, 125, 200, 275, 350, and 425 µmol m^−2^ s^−1^ at the center of each section) (Table 1). Treatments were randomly allocated to one of the six sections of each metal rack. The PPFD was controlled by sending a pulse width modulation signal from the datalogger to the dimmable drivers powering the light fixtures. The LED lights were on for 16 hours per day. Therefore, plants received daily light integrals of ~3.1, 7.4, 12.1, 16.2, 19.9, and 23.6 mol m^−2^ d^−1^ at the center of each section in the six treatments (Table 1).

### 4.3. Data Collection and Calculations

Mizuna and lettuce canopy images were captured twice a week throughout the growing period, using a chlorophyll fluorescence imaging setup. For the fluorescence imaging, we used a monochrome camera (CM3-U3-31S4M-CS, Chameleon3 USB3 camera, FLIR Systems, Inc., Arlington, VA, USA) with a 665 nm longpass filter (LP665 Dark Red Longpass Filter; Midopt Midwest Optical Systems, Inc., Palatine, IL, USA) attached to the lens. The camera was mounted facing downward inside of a 1.2 m × 0.6 m × 1.5 m grow tent. A blue LED panel was mounted inside the tent next to the camera to excite chlorophyll and induce fluorescence. Reemitted light from chlorophyll fluorescence was captured by the camera. Canopy images were taken biweekly on groups of nine plants. Those images were then analyzed with ImageJ software (ImageJ 1.52a, National Institute of Health, Bethesda, MD, USA) to determine the PCS. These PCS data were divided by nine to determine the PCS per plant. Those values were plotted against time and sigmoidal curves [f = a/(1 + exp(−(x−x0)/b))] were fitted (SigmaPlot 11.0, Systat software, Inc., San Jose, CA, USA). This was done for all individual treatments and replicates (R^2^ ≥ 0.99). Using the coefficients for the sigmoidal equation, the daily PCS was estimated (Microsoft Excel 365, Microsoft Corporation, Redmond, WA, USA). The daily PCS data were multiplied by the DLI received in each corresponding treatment to calculate the daily incident light per plant. By adding those daily incident light values, the total incident light on the canopy throughout the growing period was calculated.

One day before the harvest, leaf CCI and leaf anthocyanin content index (ACI) were measured using chlorophyll and anthocyanin meters (CCM-200 plus and ACM-200 plus; Apogee Instruments, Logan, UT, USA) on uppermost fully-expanded leaves. Then, the quantum yield of photosystem II (Φ_PSII_) and CO_2_ assimilation were measured using a leaf gas exchange system, equipped with a chlorophyll fluorometer (CIRAS-3 Portable Photosynthesis System: PP Systems, Amesbury, MA, USA). The corresponding PPFD level of each treatment was provided using the white LED light in the leaf cuvette during the measurements. The average leaf temperature, vapor pressure deficit, and CO_2_ concentration inside the cuvette were 24.5 °C, 1.3 kPa, and 781 µmol mol^−1^, respectively, to mimic the conditions inside the growth chamber at the time of the data collection.

At the end of the study, lettuce plants were harvested at 28 days after seeding and mizuna plants were harvested at 27 days after seeding. The total leaf area of each group of plants was measured using a leaf area meter (LI-3100 leaf area meter; LI-COR Biosciences, Lincoln, NE, USA). After drying them at 80 °C for 7 days, shoot dry weight of each group of plants was also measured. Both leaf area and shoot dry weight value were divided by nine to calculate the per plant leaf area and shoot dry weight. By dividing the leaf area by the associated PCS at harvest, the canopy overlap ratio was calculated. Specific leaf area (SLA) was calculated by dividing the leaf area of a plant by the shoot dry weight. To calculate LUE, shoot dry weight was divided by the total incident light that the plant received over the growing period.

### 4.4. Experimental Design and Statistical Analysis

The experiment design was a randomized complete block with a split-plot with one metal shelving rack as a block, 3 blocks, six PPFD levels as the main treatment, and the two crops as the split plot. The experimental unit was a group of nine plants. Regression analyses (linear, quadratic, sigmoidal, and exponential rise to maximum) were conducted with time or average PPFD of each treatment as the independent variable (SigmaPlot 11.0, Systat software, Inc., San Jose, CA, USA). Finally, multiple regression analysis (α < 0.05) was performed using SAS (SAS University edition; SAS Institute, Cary, NC, USA) with PPFD as a continuous and species as a class variable to test for PPFD and species effects on PCS, incident light over the growing period, CCI, ACI, Φ_PSII_, CO_2_ assimilation, leaf area, canopy overlap ratio, dry weight, SLA, and LUE. When there was no significant interaction between species and PPFD, the interaction term was removed from the model and the main effects of species and PPFD were used to describe the treatment effects.

## Figures and Tables

**Figure 1 plants-10-00704-f001:**
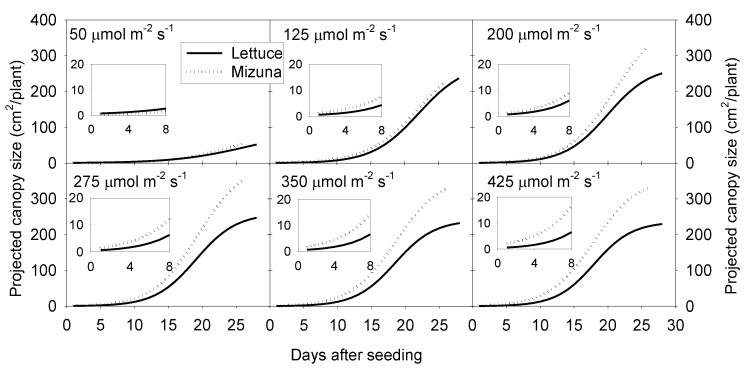
Sigmoidal regression curves fitted through the projected canopy size (PCS) data of mizuna (*Brassica rapa* var. *japonica*) and lettuce (*Lactuca sativa* ‘Green Salad Bowl’) over the course of the growing cycle for plants grown at six different photosynthetic photon flux densities (PPFD, upper left corner of each graph) (R^2^ ≥ 0.99 for all curves). Inserts show the PCS during the first eight days of the study.

**Figure 2 plants-10-00704-f002:**
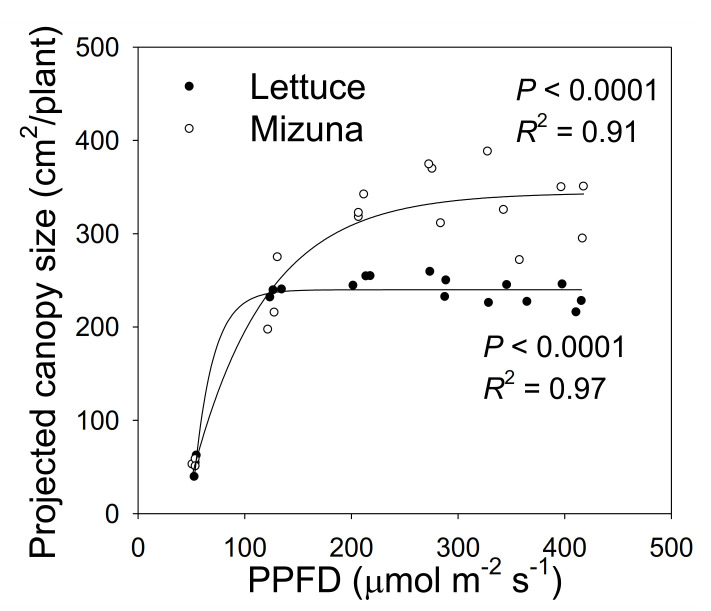
The projected canopy size (PCS) of mizuna (*Brassica rapa* var. *japonica*) and lettuce (*Lactuca sativa* ‘Green Salad Bowl’) plants grown at six different photosynthetic photon flux densities (PPFDs). Data were collected at the end of the growing cycle (27 and 28 days for mizuna and lettuce, respectively). Each data point represents the mean of nine plants.

**Figure 3 plants-10-00704-f003:**
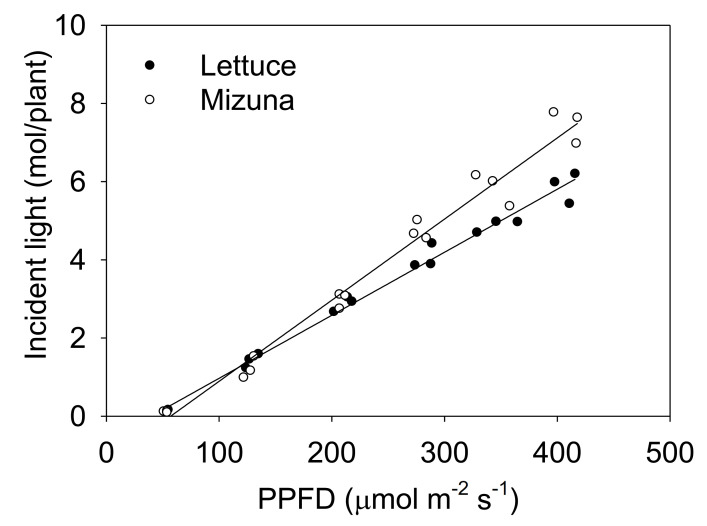
The total incident light on the plant canopy throughout the growing period of mizuna (*Brassica rapa* var. *japonica*) and lettuce (*Lactuca sativa* ‘Green Salad Bowl’) plants grown at six different photosynthetic photon flux densities (PPFDs) for 27 and 28 days, respectively. Lines show the results from multiple regression analysis (*R^2^* = 0.98), which indicated a significant species × PPFD interaction (*p* < 0.0001). Each data point represents the mean of nine plants.

**Figure 4 plants-10-00704-f004:**
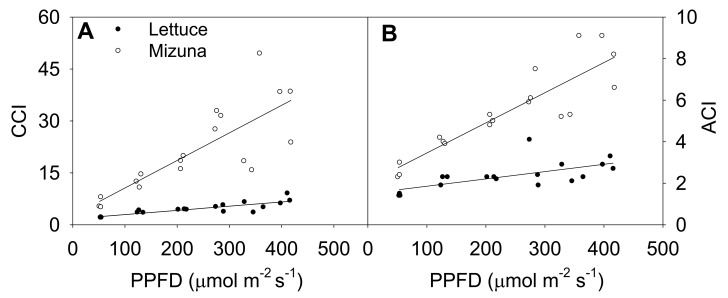
(**A**) Chlorophyll content index (CCI) and (**B**) anthocyanin content index (ACI) of mizuna (*Brassica rapa* var. *japonica*) and lettuce (*Lactuca sativa* ‘Green Salad Bowl’) plants grown at six different photosynthetic photon flux densities (PPFD). Data were collected a day before the harvesting (26 and 27 days for mizuna and lettuce, respectively). Lines show the results from multiple regression analysis, which indicated a significant species × PPFD interaction for both CCI (*R^2^* = 0.82, interaction *p* < 0.0001) and ACI (*R^2^* = 0.88, interaction *p* < 0.0001).

**Figure 5 plants-10-00704-f005:**
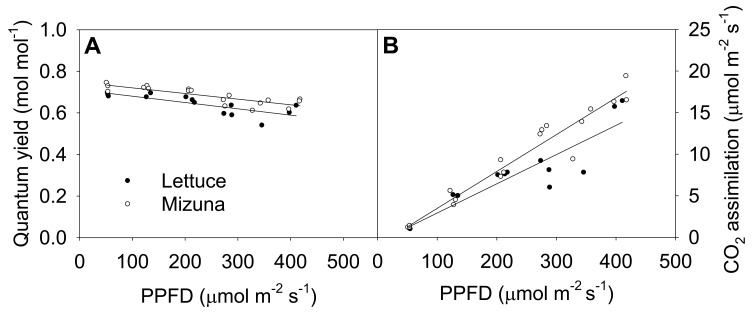
(**A**) Quantum yield of photosystem II and (**B**) net CO_2_ assimilation rate of mizuna (*Brassica rapa* var. *japonica*) and lettuce (*Lactuca sativa* ‘Green Salad Bowl’) plants grown and measured at six photosynthetic photon flux densities (PPFD*s*). Data were collected a day before the harvesting (26 and 27 days for mizuna and lettuce, respectively). Lines show the results from multiple regression analysis, which indicated no significant species × PPFD interaction for quantum yield (*R^2^* = 0.72, interaction *p* = 0.62), but significant effects of PPFD and species (both *p* < 0.0001). For CO_2_ assimilation rate, there was a weak species × PPFD interaction effect (*R^2^* = 0.91, interaction *p* = 0.08).

**Figure 6 plants-10-00704-f006:**
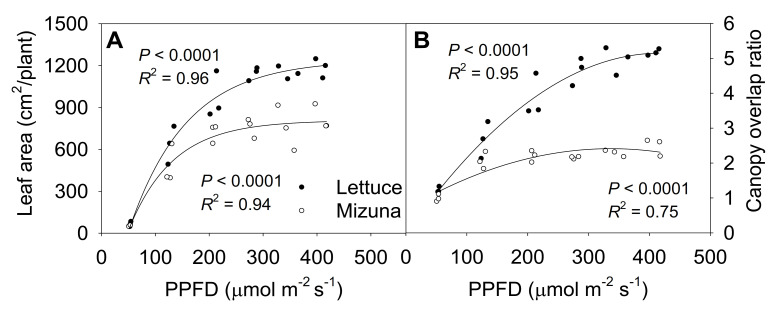
(**A**) Leaf area per plant and (**B**) canopy overlap ratio of mizuna (*Brassica rapa* var. *japonica*) and lettuce (*Lactuca sativa* ‘Green Salad Bowl’) plants grown at six different photosynthetic photon flux densities (PPFD*s*) for 27 and 28 days, respectively. The canopy overlap ratio is the ratio between the leaf area and the projected canopy size at harvest. Each data point represents the mean of nine plants.

**Figure 7 plants-10-00704-f007:**
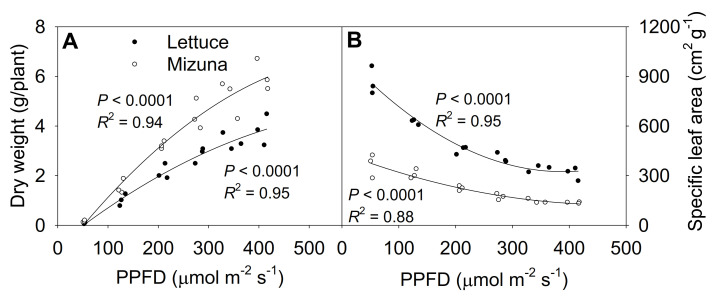
(**A**) Shoot dry weight and (**B**) specific leaf area of mizuna (*Brassica rapa* var. *japonica*) and lettuce (*Lactuca sativa* ‘Green Salad Bowl’) plants grown at six different photosynthetic photon flux densities (PPFD*s*) for 27 and 28 days, respectively. Specific leaf area is the ratio between leaf area and dry weight. Each data point represents the mean of nine plants.

**Figure 8 plants-10-00704-f008:**
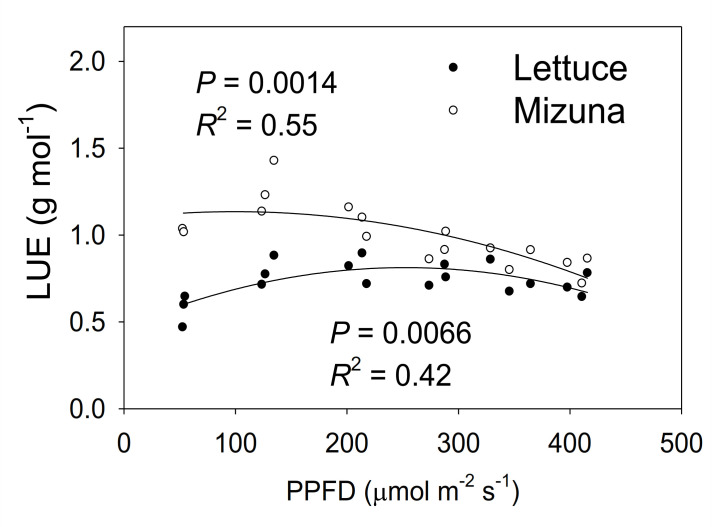
Light use efficiency (LUE, grams of shoot dry weight per mol of incident light) of mizuna (*Brassica rapa* var. japonica) and lettuce (*Lactuca sativa* ‘Green Salad Bowl’) plants grown at six different photosynthetic photon flux densities (PPFDs) for 27 and 28 days, respectively. Each data point represents the mean of nine plants.

**Table 1 plants-10-00704-t001:** Photosynthetic photon flux densities (PPFD), and the average daily light integral (DLI) at the center of the tray of each treatment. The photoperiod was 16 hours for all the treatments. Values show the mean ± standard deviation.

PPFD (µmol m^−2^ s^−1^)	DLI (mol m^−2^ d^−1^)
53 ± 2	3.1 ± 0.1
128 ± 6	7.4 ± 0.3
210 ± 9	12.1 ± 0.5
281 ± 14	16.2 ± 0.8
345 ± 15	19.9 ± 0.8
410 ± 16	23.6 ± 1.0

## Data Availability

Data used in this study is available at https://drive.google.com/drive/folders/1BgP771XRRsbZHCttrHB-SueqV4nU0JMf.

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
