# Peer review of "Canopy Size and Light Use Efficiency Explain Growth Differences between Lettuce and Mizuna in Vertical Farms"

_plants, 2021, doi:10.3390/plants10040704_

Round 1

Reviewer 1 Report

This manuscript describes a study on the effects of different photosynthetic photon flux densities on the canopy size, biomass, and light use efficiency of lettuce and mizuna grown under sole-source lighting. The authors concluded that canopy size and light use efficiency can predict crop biomass and can be used to screen fast-growing crops suitable for vertical farming. The manuscript is well written and clearly structured. The experiment design was robust. The data collected were sufficient and appropriate to address research questions. However, the authors should improve justification of the study and the relevance of the results to vertical farmers. There are some graphing, grammatical, and other errors across the manuscript. Please find detailed comments below.

Line 9: Please change “the energy costs…are high” to “the energy cost…is high” to be consistent with the following sentence.

Line 11: Please change “crop” to “crops”.

Line 14: Please change “16 h/day” to “16 h d-1” to be consistent.

Line 14: Is “biomass” fresh or dry mass, or both? Please change “in” to “than” and add how long the crop cycle was, and whether mizuna and lettuce were harvested at the same time. Define “higher PPFDs” with a specific range.

Line 15: “Because of” – correlation does not equal causation. Net photosynthesis should be discussed here before claiming canopy size as the cause of growth differences. Please add “the” before “projected”.

Line 17: Please delete “a”.

Lines 20 to 23: It is well known that whole plant photosynthesis is the product of net photosynthesis and specific leaf area. The conclusions of this study agree with this notion. Please justify the novelty and necessity of the study and its unique contributions.

Line 33: Please change “is” to “are”.

Lines 37 to 39: Reference(s) are needed here.

Line 43: Please add “until a saturation PPFD is reached” or similar.

Line 46: This is the first time PPFD appears. Please define it here, and not later in the text.

Line 55: Please be consistent about whether to spell out abbreviations at the start of a sentence. In Line 51, PCS is spelled out, but here, LUE is not spelled out.

Line 57: Please change “crop” to “crops”.

Line 77: Please define “CCI”.

Lines 79 to 86: This justification is not convincing. First, screening for rapidly growing plants in vertical farms can be done by growing different crops and comparing their growth rates (yield/time) without knowing the exact underlying causes. Second, the literature summarized in the Introduction has already shown plants with higher PCS and higher LUE are more productive, so why is this knowledge insufficient to explain the growth differences between mizuna and lettuce in the previous study? In other words, why was this study warranted? The fact that PCS and LUE data were not previously collected is not convincing justification.

Lines 88 to 89: The justification of why different PPFDs were evaluated needs support since numerous studies have already investigated the effects of PPFDs on crop morphology and physiology. Please be specific about the knowledge gap this study aimed to fill, and how this study was novel.

Lines 92 to 93: This is a repetitive sentence to lines 83 to 84 and should be removed.

Line 94: Please change “~” to “approximately” or “≈”.

Line 97: CCI should have been defined previously at first appearance.

Line 99: Please delete “at”.

Line 100: Please abbreviate LUE.

Lines 113 to 114: This sentence can be misleading. It can mean the PCS of mizuna was similar to the PCS of lettuce, which is not the case. Please rephrase it more clearly.

Line 120: Please provide the sample number and sampling frequency (seems to be twice a week), mark actual data points on the charts (should not be continuous), and state if the curves are smoothed. Also, please check all the ticks (e.g., revise the left ticks in the 125 umol chart, add right ticks in the 200 umol chart, etc.).

Line 122: For at least Figures 2 and 3, there are three data points around each PPFD level for each crop. However, the materials section says there were nine plants per crop in each treatment. Please clarify if the data points are for three replicates, each representing nine plants. Also, why not make this chart for every week to see the relationships between PPFD and PCS over time?

Lines 128 to 129: Please conduct mean separations to state whether trends are different (and at point). It seems data of the two crops were not different at 200 umol.

Line 134: Please change “PPFD” to “PPFDs” here and elsewhere.

Line 137: Why measure anthocyanin content? Isn’t lettuce ‘Green Salad Bowl’ green? What is the relevance of ACI data to green-leaf crops?

Line 151: Please make sure the x axis includes 500 umol so all data in Figure 4A are within the described range (also check other charts for this).

Lines 158 to 159: Is this difference statistically significant based on mean separations? Please refrain from drawing conclusions only based on the visual or numerical distance between linear regression lines. “Always” is not appropriate here. It seems at some PPFD levels, the quantum yield is similar between the two crops.

Line 168: Please add plant ages when the data were collected.

Line 195: The unit of the y axis for Figure 7B should be cm2 g-1. Also, the lines between lettuce and mizuna are converging as PPFD increased to around 425 umol. Please describe this in the text.

Lines 213 to 214: This is a repeating sentence to what has been stated before and should be removed.

Lines 227 to 229: Please provide the context of the lighting conditions for this result (e.g., a fixed white spectrum at a certain photoperiod).

Line 315: Please correct the spelling of “photosynthesis.”

Line 326: Please change “maintenance” to “maintain”.

Line 331: What is the “relative growth rate” referred to here? Why is it lower for larger plants that have higher yields (growth rates)?

Line 341: What is the approach referred to here? Measuring canopy size and LUE with specialized equipment? For typical vertical farmers, is this realistic? Why not just compare yields of different crops to screen fast-growing crops? How is phenotyping mentioned later relevant for typical vertical farmers to select crops?

Line 333: Does this study suggest mizuna should be selected over lettuce because of its higher canopy size and LUE since this study is positioned to find a way to select fast-growing crops to increase LUE? There are many leafy green crops that grow faster than lettuce, but lettuce is still the most popular crop in indoor farming. Please address the significance of the results and provide the context that crop selection is a result of multiple factors, and not just growth rates.

Reviewer 2 Report

Using PCS and LUE to evaluate growth potential of a crop under a vertical farming setting looks interesting to a wide range of the readership. On top of that, the LUE concept and its relationship to PCS used in the study definitely provides an insight to the CEA research communities that are prone to use the growing-space-based LUE and actual leaf area values for estimating the light interception. The experiment was well-designed and the conductance of the corresponding data analysis was sound. There are few points to be clarified or addressed for publication. Please take a look at those in the comments in the attached file.
